# Exposure to organochlorine pesticides as a predictor to breast cancer: A case-control study among Ethiopian women

**Seblework Mekonen**[1]*, **Mohammedgezali Ibrahim**[1], **Higemengist Astatkie**[1], **Aynalem Abreha**[2]

**1** Department of Environmental Health Science and Technology, Faculty of Public Health, Jimma University, Jimma, Ethiopia, **2** Department of Oncology Addis Ababa, College of Health Sciences, University, Addis Ababa, Ethiopia

* seblework2001@yahoo.com

## Abstract

Breast cancer (BC) is becoming one of the most prevalent non-infectious disease in low and middle income countries. The steady rise of BC incidence may be related to the different risk factors. Among many, rampant presence of environmental pollutants might be one of the risk factors. Therefore, the aim of this study is to investigate exposure to organochlorine pesticides as a risk factor to breast cancer. A case-control study design was employed among breast cancer patients and non-breast cancer individuals (controls). Blood samples were collected from 100 study participants (50 cases and 50 controls) followed by serum separation, extraction and cleanup using standard analytical procdures. The findings revealed that ten organochlorine pesticides were detected in the serum of the study participants. From the detected organochlorine pesticides, heptachlor was observed at higher concentration for breast cancer patients (6.90±4.37 μg/L) and controls (9.15±3.84 μg/L). Mean serum level of p,p'-DDE, p,p'-DDT, heptachlor, gamma-chlordane, endosulfan, and dibutyl-chlorendate were significantly higher in the serum of breast cancer patients than the controls. From the studied pesticides, p,p'-DDT and gamma-chlordane are significant predictors for BC, while, others are equivocal. A unit increment of the concentration of p,p'-DDT (AOR; 2.03, 95% CI: 1.041–3.969) increased the odds of developing breast cancer by two, while for gamma-chlordane (AOR;3.12, 95% CI; 1.186–8.203) by three. Our study results suggesting that, organochlorines are a risk factors for breast cancer in Ethiopia. Decreasing exposure to such organochlorines might have a significant public health relevance in reducing non-communicable chronic illnesses. Besides, continues monitoring of persistent organic pollutants using body biomarkers is important for disease prevention and device mitigation measures.

**Data Availability Statement:** Data for the concentration of Organochlorine pesticides in serum of study participants were submitted with the paper as Supporting Information.

## Introduction

Breast cancer is recognized as one of the top cancer cases among women worldwide. It is increasing from time to time particularly in low income countries where most cases are

**Funding:** The grant holder is Seblework Mekonen. This study was carried out by the aid of a grant obtained from the United Nations Educational, Scientific and Cultural Organization (UNESCO) and International Development Research Centre (IDRC) Ottawa Canada. Award agreement No.: 4500384863. The funders had no role in study design, data collection and analysis, decision to publish, or preparation of the manuscrip.

**Competing interests:** The authors' declare that they have no computing interest

diagnosed in late stages and due to the different risk factors associated with the development of the disease [1, 2]. In the year 2011, it is estimated that over 508,000 women died due to breast cancer [3]. Even though breast cancer is considered as a disease of the developed nations, almost 50% of breast cancer morbidity and 58% of mortality occur in developing countries [4]. According to Jemal and his colleagues [5], about 56% of cancer cases and more than 60% of deaths are from newly diagnosed breast cancer cases in developing countries.

Breast cancer has several modifiable and non-modifiable risk factors. The non-modifiable risk factors are age, sex, genetic factor, family history of previous breast cancer cases and modifiable risk factors such as menstrual and reproductive factors, radiation therapy, hormonal therapy and some environmental pollutants such as smoking, and other chemicals [6]. Organochlorine pesticides are used as an insecticide, fungicides, and termiticides worlsd wide. These pesticides have the property of slow degradability, lipophilicity, able to bioaccumulate, and biomagnify in a food chain. As a result, they may be detected in human adipose tissue, blood and breast milk [7]. According to the study done in Ethiopia [8], organochlorine pesticides were detected in breast milk collected from mothers. Therefore, the rising of pesticide application and contamination of food items, and body accumulation might be one of the risk factors for breast cancer. From the animal studies organochlorine chemicals have caused various cancers [9]. However, epidemiological studies regarding environmental pollutants such as chlorinated pesticides and dietary related problems are lacking [10].

Different studies hypothesized the association between organochlorine pesticides and cancer cases, however, well established risk factors explain only about 41% of breast cancer causes. There are also unidentified factors cuasing breast cancer. This has attracted different researchers to investigate organochlorine pesticides and other environmental pollutants as possible causes of breast cancer [4]

From a study done by Turusov and his colleagues [11], DDT and DDE can disrupt the endocrine system and concluded as they are possible carcinogens in humans.

The problem could be severe in countries like Ethiopia where pesticide control is loos and organochlorines pesticides, such as DDT and others, are in use for a more extended period of time. This is known to contaminate the environment and cause health risks. Moreover, from a study done in southwestern Ethiopia, most of the staple food items and drinking water were contaminated with organochlorine pesticides [12–14] which can able to accumulate in consumers body. These all may lead to the exposure of human and resulted in chronic diseases including breast cancer. Records of different hospitals in Ethiopia indicate that more than 150,000 cancer cases per year, contributing to 4% of all deaths. According to the report of the cancer registry in Addis Ababa Ethiopia, 34% of all female cancer cases are breast cancer cases [15].

In addition, McPheron and colleagues [16], mentioned that, from the known cancer cases, only half could be attributed to established risk factors such as null parity and late age at first childbirth and others might be from Environmental pollutants including pesticides. Recognizing all these risk factors, about 70% of females who develop breast cancer do not have identifiable risk factors [17]. However, cancer is the most neglected health issues in Ethiopia [18]. Even though, there are different risk factors of breast cancer, the cause of increasing prevalence of this disease in Ethiopia and many low income countries is not known. Moreover, data on the relationship of Organochlorine pesticides (OCPs) with breast cancer is scarce, which might be one of the factors in developing countries. Therefore, the present study aimed at investigating the organochlorine pesticides residue among breast cancer patients and healthy individuals as risk factor for breast cancer.

## Materials and methods

### Study setting and period

The study was conducted in Addis Ababa University, Oncology Unit of Black-Lion Specialized Hospital (BLSH), Ethiopia, from February to April 2020. BLSH is a referral and teaching hospital of Addis Ababa University in Addis Ababa, Ethiopia. The hospital is the only government-owned site for comprehensive cancer diagnostic and treatment centers within the country. **Fig 1** indicates a map of the study area.

### Study design

A case-control study was conducted to determine the pesticide residues in serum of pathologically diagnosed breast cancer patients and non-breast cancer control groups. The design aimed at evaluating the association between the level of OCPs in serum and breast cancer risk among the study groups. Patients of young adults to old adults (18 to 55+ years age groups) visited the oncology unit of BLSH during the study period, and those who were volunteer to participate in the study were included as the study population. Cases were women diagnosed with breast cancer, while controls were patients or caregivers in a similar age group without breast cancer.

### Data collection

Data were collected by interviewing study participants (breast cancer patients, patient relatives, or caregivers) using semi-structured questionnaires and trained interviewers after getting their written consent. The semi-structured questionnaires were focused on the assessment of exposure to OCPs as risk factors to breast cancer, socio-demographic characters of the participants,

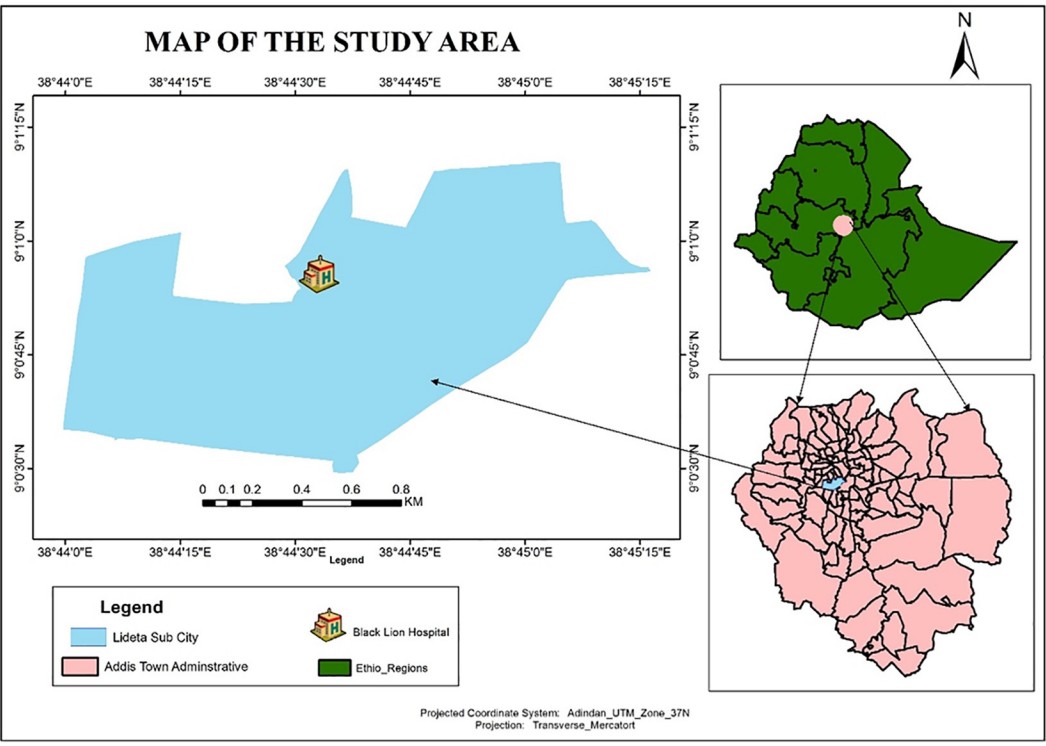

**Fig 1. Map of Addis Ababa and the location of black lion specialized hospital, in Addis Ababa, Ethiopia.**

medical history, and occupational exposure to pesticides. The designed questionnaires were translated into the local language before interviewing the participants.

## Blood sample collection and serum separation

A total of 100 blood samples were collected using a voluntary based convenient sampling technique. Based on their written informed consent, 5 mL of venous blood samples were collected by well-experienced nurses using sterile syringes and transferred to serum separator tube (SST) after the cleansing of the venipuncture site with a swab soaked in methylated spirit. Then the blood samples were allowed to clot for 30 minutes and consecutively centrifuged at 3500 rpm for 5 minutes to separate the serum and serum separation was done by well experienced medical laboratory professionals. The serum was transferred into Nunc™ Tube immediately after separation, and the samples were stored at -20˚C until transported for further analysis. The storage temperature was checked every six-hours, and the serum samples were transported to Jimma University Environmental Health Laboratory using iceboxes with an icepack.

## Extraction and clean up

The extraction and clean-up of the serum samples were done using analytical methods as described by Attaullah et al. and Turci et al. [19, 20] with minor modification. The procedures weere described as follows. An aliquot of 2 mL serum was extracted first using 1 mL of methanol (99.8%) and vortexed for 30 s. Then 5 mL diethyl ether (99%) mixed with (1:1, v/v) n-hexane (99%) and agitated for 1 min then centrifuged for 5 min at 2500 rpm. The upper organic phase was collected, and the aqueous phase was extracted again with the same solvents. The upper layer was taken and combined with the previous organic phase. The combined organic extracts were cleaned up/purified through glass Column Chromatography containing 60–100 mesh size Florisil (1 g) and topped with 1 g of the UK made; code S/6840/53 anhydrous Sodium Sulfate. The column was rinsed by n-hexane before the addition of Florisil and anhydrous Sodium Sulfate. The extracts were eluted with 3 mL n-hexane followed by 3 mL n-hexane/diethyl ether (1:1, v/v). Then elute evaporated to dryness at 40˚C using a Germany made (Heidolph S.No: 020815801) rotavapor. Then the concentrate was dissolved into 2 mL of n-hexane and transferred to a 1.5 mL crimp neck vial for Gas-chromatography with Electron Capture detector (GC-ECD) analysis.

## Standard preparation and linearity of the standard curves

The stock solution was prepared by dissolving 50 mg of each standard pesticide powder in the solvent of choice for each organochlorine pesticide. An intermediate solution was prepared by taking 1 mL of stock solution and diluting it in 9 mL of the corresponding solvent. Working standard solutions were prepared at concentration ranges of 0.001 to 10 mg/l by serial dilution of the intermediate solution. The linearity of the standard pesticides was measured at five different concentrations of 10, 1, 0.1, 0.01, and 0.001 mg/l from the mixture of all standards of their working solutions. The identification of the target pesticides were done by running known concentrations for each pesticides and looking at the retention time. Calibration curves were plotted by taking concentration against the peak area of the individual pesticides, and the determination coefficient $r^2$ is greater than 0.999 for all of the pesticides under study.

## Chromatographic conditions

Agilent Technologies 7890A GC-ECD was used for the identification and quantification of organochlorine pesticides in human serum. Nitrogen gas generator with 99.99% purity was

used as a carrier and makeup gas. An analytical column HP5, No 19091J-413 with a dimension of 30 m x 0.25 μm film thickness x 0.32 mm internal diameter was used following an oven temperature program similar to the study done by Mekonen and her colleagues [12], for the determination of OCPs. The initial temperature of the oven was set at 80˚C, ramp at a rate of 30˚Cmin$^{-1}$ to 180˚C, then ramp at a rate of 3˚C min$^{-1}$ to 205˚C for 4 min holding time, and then ramp at a rate of 20˚C min$^{-1}$ to 290˚C and holding for 8 min. The total run time was 27.92 min. A volume of 1 μL aliquot was injected in splitless mode. The pesticide residues were detected with μ-ECD detector operating at a temperature of 300˚C and using nitrogen as a makeup gas at a flow rate of 60 mL/min. The samples were analyzed in triplicate, and the mean concentration was computed accordingly.

## Quality control

In order to assure the data quality, the questioners were translated to the local language before data collection and was pre-tested for evaluation of the data collection tool. Both blood sample collection and questionnaire survey data collections were done by experienced health professionals and trained data collectors, respectively. The training was given to the data collectors to have a common understanding of the data collection tools and study variables. Laboratory personnel was blinded with respect to the case or control status of the study participants during sample analysis. The samples were labeled and coded accordingly to avoid confusion of results and stored at an appropriate temperature. Maximum care was taken for the study participants and for samples starting from collection, transportation, storage, and instrumental analyses to estimate pesticides in the serum samples.

## Data bases and statistical analysis

A Data base was established by enumerating laboratory results along with the data's from occupational and socio-demographic characteristics. The data was cleaned and checked for completeness manually. Epi Data 3.1 was used for data entry, and the data were exported to SPSS V, 23 for further analysis. Descriptive statistics were used to compute socio-demographic, biological, and behavioral characteristics. Independent sample t-test for risk factor profile and Mann-Whitney U test for serum level of OCPs was used to compare the significant difference between the two study groups (cases and controls).

The association between breast cancer and each covariate were assessed first by bivariate logistic regression and Chi-square test for continuous and categorical variables to identify candidate variables for the multivariate logistic regression model. Variables with P-value < 0.25 were taken to multivariable Logistic regression. The odds ratio was estimated with 95% CI to show the strength of association and P-value< 0.05 was used to declare statistical significance. The goodness of fit of the final model was checked using Hosmer and Lemeshow test of goodness of fit and considering good fit at P-value ≥ 0.05. Logistic regression is a model that can give us an odds ratio which is controlled for multiple confounders. This odds ratio is known as the adjusted odds ratio, because its value has been adjusted for the other covariates (including confounders) but in our study prior to adjusted odds ratio we have analyses the crude odds ratio for the entire predictor variables with response variable each at a time. Based on the crude odds ratio statistical significance those predictor variables which were significant were taken to the adjusted odds ratio. But some of the Organochlorine variables especially those with many non-detected values mean that those founded in some of the respondents only were removed from the model eg. Endosulphan. So the adjusted odds ratio indicates that the effects of one predictor given that is adjusted for the effects of others. The value associated with odds ratio can be categorized into three. "OR = 1" means there is no association between

the predictor and the response variable, "OR<1" MEANS the given predictor variable is protective for the response variable for further illustration, the odds of exposure among case-patients are lower than the odds of exposure among controls. The exposure might be a protective factor against the occurrence of the disease by associated value with odds ratio.

## Ethical considerations

Ethical clearance was taken from the ethical review board of Jimma University, Institute of Health before prior to data collection. Besides, ethical clearance was obtained from the institutional review board (IRB) of Addis Ababa University, College of Health Sciences Ethiopia. Maximum care was taken to not harm the study participants. Before the blood sample collection, the purpose of the study was clearly explained to the study participants. Breast cancer patients and control groups were asked individually for their willingness to participate in the study using written consent. They were interviewed and blood sample collection was undertaken whenever we get their consent. Personal identifiers of study participants will not be used for reporting, publication, and related activities. On the other side, study participants with higher pesticide residue levels were personally contacted for further intervention. The study was to take the utmost care not to contaminate the environment and not expose the blood sample collectors, workers and technicians in the field and the laboratory.

## Results

### Socio-demographic characteristics of study participants

The mean age of cases and controls were 43.06 (±7.89) and 39.44 (± 8.86) years, respectively. The difference in the mean age of the two groups was statistically significant ($P = 0.034$). Regarding the educational status, 23 (46%) and only 10 (5%) of the cases and controls had secondary and higher educational levels, respectively, while 15 (30%) of the cases and 26 (52%) of the controls have no formal education. Most of the breast cancer patients 43 (86%) were living in an urban areas, while others were from rural sites. Most of the participants in cases and controls are married 34 (68%) and 44 (88%), respectively. Regarding the occupational status of the study participants about 21 (42%) of the breast cancer patients are government employees, while 23 (46%) of controls are farmers (**Table 1**).

**Table 1. Socio-demographic characteristics of study participants.**

| Variables | | Study participants | | Total |
|---|---|---|---|---|
| | | **Case** | **Control** | |
| Educational status | No formal education | 15 (30%) | 26 (52%) | 41 (41%) |
| | Primary | 12 (24%) | 19 (38%) | 31 (31%) |
| | Secondary and above | 23 (46%) | 5 (10%) | 28 (28%) |
| Residencial area | Rural | 7 (14%) | 21 (42%) | 28 (28%) |
| | Urban | 43 (86%) | 29 (58%) | 72 (72%) |
| Occupation of respondents | Farmer | 12 (24%) | 23 (46%) | 35 (35%) |
| | Merchant | 2 (4%) | 3 (6%) | 5 (5%) |
| | Employee | 21 (42%) | 4 (8%) | 25 (25%) |
| | House wife | 15 (30%) | 20 (40%) | 35 (35%) |
| Marital status | Single | 6 (12%) | 1 (2%) | 7 (7%) |
| | Married | 34 (68%) | 44 (88%) | 78 (78%) |
| | Divorced | 5 (10%) | 3 (6%) | 8 (8%) |
| | Widowed | 5 (10%) | 2 (4%) | 7 (7%) |

## Risk factors status of study participants

Regarding the smoking status, 94% of the study population were nonsmokers, and about 21 (42%) of breast cancer patients and 26 (52%) of the controls were experienced alcohol drinking. Regarding contraceptive use, more than half of the breast cancer patients 31 (62%) and controls 29 (58%) were used one or more types of contraception methods. From the study participants, about 98% of them had regular menses at the age of 18. From the participants, 95.7% breastfed their children for an average of 22.8±6.5 and 23.9±6.2 months for cases and controls, respectively. Most of the study population (91%), the commonly used food items were plant products such as Teff, wheat, vegetables, and fruits which may be a source for the accumulation of OCPs in their body and only 6 (12%) of the breast cancer patients had a familial history of cancer and most of the cases are not inherited.

The mean age at menarche was 14.88 ±1.38 and 15.28 ±1.05 years for breast cancer patients and controls, respectively. There is no significant difference (p-value>0.05) in the biological/reproductive risk factors between the breast cancer patients and the controls. However, a statistically significant difference was observed in the mean number of children between cases (3.74±1.9) and controls (5.16±2.2) (P = 0.001). The mean age at first birth for cases were (22 ±4.11) and controls (20.3±2.99) the findings shows a significant difference (P = 0.029). The risk factors profile of study participants presented in **Table 2**.

## Pesticide usage and serum concentration of Organochlorine Pesticides (OCPs)

From the survey results, 15 (30%) and 26 (52%) of cases and controls had experienced domestic agricultural farming. Concerning pesticide usage, 12 (24%) of the cases and 26 (52%) of

**Table 2. Status of the biological and behavioral risk factors of the study participants.**

| Variables Status | | Mean | S.D | *P*-Value |
|---|---|---|---|---|
| Age of respondents (years) | Case | 43.06 | 7.89 | 0.034 |
| | Control | 39.44 | 8.86 | |
| Age at first menses (years) | Case | 14.88 | 1.38 | 0.106 |
| | Control | 15.28 | 1.05 | |
| Age at menopause (years) | Case | 44.41 | 2.59 | 0.348 |
| | Control | 45.11 | 1.67 | |
| Number of children | Case | 3.74 | 1.90 | 0.001 |
| | Control | 5.16 | 2.20 | |
| Age at first birth (years) | Case | 22.00 | 4.11 | 0.029 |
| | Control | 20.31 | 2.99 | |
| Duration of breastfeeding(months) | Case | 22.80 | 6.50 | 0.413 |
| | Control | 23.90 | 6.20 | |
| No of miscarriage or abortion | Case | 1.57 | 0.75 | 0.215 |
| | Control | 1.28 | 0.46 | |
| Duration of contraceptives usage (years) | Case | 4.29 | 4.55 | 0.125 |
| | Control | 3.06 | 1.93 | |
| No of cigarettes/day | Case | 10.67 | 1.15 | 0.398 |
| | Control | 14.00 | 6.00 | |
| Hours spend in cigarettes smoke area | Case | 1.00 | 0.00 | 0.321 |
| | Control | 3.00 | 2.65 | |
| Age at habitual drinking (years) | Case | 18.33 | 3.54 | 0.113 |
| | Control | 16.50 | 4.10 | |

**Table 3. Serum concentrations of Organochlorine Pesticides (OCPs) in the study population.**

| Types of OCPs | Median (µg/L), Mean ± SD (µg/L) | | P-Value |
| --- | --- | --- | --- |
| | Cases | Controls | |
| o,p'-DDT | 1.58, 1.86 ± 1.07 | 2.16, 2.64 ± 1.56 | 0.301 |
| p,p'-DDD | 0.98, 1.58 ± 1.20 | 0.93, 1.15 ± 0.82 | 0.252 |
| p,p'-DDE | 1.12, 1.99 ± 2.24 | 1.93, 2.32 ± 1.75 | 0.045 |
| p,p'-DDT | 2.49, 2.72 ± 1.44 | 1.59, 1.73 ± 0.82 | 0.008 |
| Total DDT | 3.80, 5.20 ± 4.84 | 3.77, 4.37 ± 2.82 | 0.976 |
| heptachlor | 7.20, 6.90 ± 4.37 | 10.37, 9.15 ± 3.84 | 0.009 |
| aldrin | 0.64, 1.20 ± 1.13 | 0.71, 1.45 ± 1.41 | 0.573 |
| heptachlor-epoxide | 1.63, 1.79 ± 0.89 | 1.47, 1.76 ± 0.93 | 0.695 |
| gamma-Chlordane | 1.38, 1.58±1.10 | 0.69, 0.96 ± 0.93 | 0.003 |
| endrin | 4.13, 3.44 ± 0.95 | 0.77, 1.75 ± 2.22 | 0.151 |
| endosulfan | 1.16, 1.33 ± 0.65 | 4.33, 3.72 ± 2.04 | 0.029 |
| dieldrin | 2.38, 2.38 ± 2.38 | 3.32, 3.33 ± 2.21 | 0.643 |
| methoxychlor | 2.78, 3.19 ± 1.33 | 3.15, 2.69 ± 1.38 | 0.337 |
| dibutyl-chlorendate | 0.74, 1.12 ± 1.17 | 1.95, 2.12 ± 1.30 | 0.017 |

OCPs = Organochlorine Pesticides, SD = Standard deviation, Total DDT = ∑ p,p'-DDT, p,p'-DDE, p,p'-DDD and o,p'-DDT.

controls had applied pesticides on their farming areas from those applied pesticides 38 (76%), and 46 (92%) of the study participants in cases and controls had applied insecticides in their dwellings, respectively. On the other hand, 95.1% of cases and 97.8% of controls did not use personal protective equipment (PPE) during pesticide applications and this will lead to exposure to the applied pesticides. None of the breast cancer patients and controls had attained pesticide-related training.

From all of the OCPs detected in human blood serum, heptachlor is observed with higher mean concentration in cases (6.90±4.37 µg/L) and controls (9.15±3.84 µg/L). While, dibutyl-chlorendate in breast cancer pataints (1.12±1.17 µg/L) and gamma-chlordane in controls (0.96 ±0.93 µg/L) are detected in lowest mean concentrations. The difference in mean serum level of OCPs between the breast cancer pataients and controls is statistically significant for p,p'-DDE (P = 0.045), p,p'-DDT *(P = 0.008)*, heptachlor *(P = 0.009)*, gamma-chlordane *(P = 0.003)*, endosulfan *(P = 0.029)* and dibutyl-chlorendate *(P = 0.017)* and the results are presented in **Table 3**.

## Association of serum level of OCPs with breast cancer

The individual organochlorine pesticides association with breast cancer was determined by using bivariate logistic regression. p,p'-DDT *(P = 0.004)* (COR = 2.231, (95% CI, 1.283–3.880)), heptachlor *(P = 0.011)* (COR = 0.877, (95% CI, 0.792–0.970)), gamma-chlordane *(P = 0.019)* (COR = 1.977, (95% CI, 1.117–3.500)), and dibutyl-chlorendate *(P = 0.033)* (COR = 0.427, (95% CI, 0.236–0.942)) were showed statistically significant association with breast cancer (**Table 4**).

Considering the effects of confounders of breast cancer and the coincidence of OCPs, as a rule of thumb, variables with a level of significance greater than 0.25, were considered for candidate selection for the multivariate logistic regression model. In addition to the above mentioned significant OCPs, endrin *(p = 0.156)*, o,p'-DDT *(p = 0.234)*, p,p'-DDD *(P = 0.140)* and endosulfan *(P = 0.070)* were the potential candidates for final regression model. However, due to very low percent detection, endrin (10% in cases and controls), o,p'-DDT (22% in cases,

**Table 4. Bivariate logistic regression results of OCPs and breast cancer association.**

| Types of OCPs | (COR) | 95% CI | P-value |
|---|---|---|---|
| o,p'-DDT | 0.602 | 0.261–1.388 | 0.234 |
| p,p'-DDD | 1.556 | 0.865–2.797 | 0.140 |
| p,p'-DDE | 0.916 | 0.728–1.153 | 0.359 |
| p,p'-DDT | 1.698 | 1.283–3.880 | 0.019 |
| Total DDT | 1.058 | 0.946–1.184 | 0.323 |
| heptachlor | 0.877 | 0.792–0.970 | 0.011 |
| aldrin | 0.859 | 0.581–1.270 | 0.446 |
| heptachlor-epoxide | 1. 036 | 0.665–1.614 | 0.378 |
| gamma-Chlordane | 1.977 | 1.117–3.500 | 0.019 |
| endrin | 1.916 | 0.781–4.704 | 0.156 |
| endosulfan | 0.286 | 0.074–1.109 | 0.070 |
| dieldrin | 0.777 | 0.331–1.825 | 0.563 |
| methoxychlor | 1.350 | 0.696–2.621 | 0.375 |
| dibutyl-chlorendate | 0.472 | 0.236–0.942 | 0.033 |

12% in controls), and endosulfan (18% in cases, 10% in controls) were removed from the multivariate model.

In the final model, after adjustment for confounders of breast cancer such as age, contraceptive use, smoking, alcohol drinking and others, p,p'-DDT ($P = 0.038$) (AOR;2.03, 95% CI; 1.041–3.969) and gamma-chlordane ($p = 0.021$) (AOR;3.12, 95% CI; 1.186–8.203) were significantly associated with increased risk of breast cancer with an adjusted odds ratio of 2.03 and 3.12, respectively. Multivariate logistic regression output is presented in **Table 5**.

This indicates that the increment in a unit concentration of p,p'-DDT, and gamma-chlordane increases the odds of developing breast cancer by 2.03 and 3.12 times, respectively. Therefore, the Serum levels of p,p'-DDT and gamma-chlordane can be an independent contributing factor for breast cancer development and supports the hypothesis of this study.

## Discussion

From the findings of our study the orgochlorine pesticides (OCPs) such as DDT and its metabolites (p,p'-DDE, p,p'-DDD, p,p'-DDT and o,p'-DDT), heptachlor, heptachlor-epoxide, Aldrin, gamma chlordane, dibuty-chlorinedate and methoxychlor are detected in the serum of the study participants. The presence of both the parent and metabolites of OCPs in the serum of the study participants may be due to the current and historical use of organochlorine pesticides in Ethiopia. The exposure of the study participants to OCPs could be from occupational activities, residential areas, or from consumption of pesticide contaminated food and drinking water as reported in different studies [12, 13, 21, 22]. According to the Stockholm convention, OCPs are a group of chemicals that are considered as Persistent Organic Pollutants (POPs)

**Table 5. Findings from multivariate logistic regression model.**

| OCPs Group | (COR) | (AOR) | 95% CI | P-value |
|---|---|---|---|---|
| p,p'-DDT | 2.23 | 2.03 | 1.04–3.97 | 0.038 |
| gamma-Chlordane | 1.98 | 3.12 | 1.19–8.20 | 0.021 |
| heptachlor | 0.88 | 0.78 | 0.59–1.02 | 0.067 |

COR = Crude Odds ratio, AOR = Adjusted Odds ratio CI = Confidence interval.

[23] which continues to contaminat our environment and leading to different chronic health risks to human. Organochlorine pesticides have been known as exogenous or xenoestrogens capable of initating or mimicking the action of hormones and changes in growth factors that may result in carcinogenicity [24, 25]. Such properties of the OCPs could be one of the risk factors for breast cancer in Ethiopia where our study tries to indicate. The concentrations of heptachlor (6.90 ± 4.37, 9.15 ± 3.84 μg/L), methoxychlor (3.19 ± 1.33, 2.69 ± 1.38 μg/L), aldrin (1.20 ± 1.13, 1.45 ± 1.41 μg/L), and dieldrin (2.38 ± 2.38, 3.33 ± 2.21 μg/L) were detected in the serum of cases and controls, respectively. The average blood serum concentration of the OPCs are higher in breast cancer pataints than the controls. Our result is supported by a study done in India, which revealed that the mean blood serum concentration of OCPs were higher in breast cancer cases than controls [26].

From our study results, heptachlor and heptachlor-epoxide were the most prevalent pesticides detected in the serum of cases and controls. Besides, from all detected OCPs, the highest mean serum concentration detected was heptachlor (6.90 ± 4.37 μg/L) for cases and controls (9.15 ± 3.84 μg/L). This is probably due to the historical and recent use of these highly chlorinated pesticides for agricultural and public health purposes in the country.

However, heptachlor and heptachlor epoxide were not associated with the increasing risk of breast cancer. Rather, heptachlor showed protective results before the adjustment of the confounders' (OR = 0.877, 95% CI, 0.792–0.970, $P = 0.011$). On the contrary, Kaur and his collegues reported that [26] that, heptachlor was highly associated with an increased risk of breast cancer in India. This might be due to the concentration of heptachlor observed in our study is not high enough to be a predicting factor for breast cancer in Ethiopia. In addition, it could be due to heptachlor and its metabolite possess a low chronic risk to humans [27].

The findings of the present study revealed that from metabolites of DDT, p,p'-DDE was one of the frequently detected pesticides (72% and 90%) in cases and controls, respectively. This may be due to the long term or historical use of DDT for agriculture as well as public health issues in Ethiopia. Higher percent detection of p,p'-DDE was also observed in controls than breast cancer cases. This could be due to many of the participants in controls were came from rural areas may exposed from consumption of different foods and may be exposed agricultural applications as most of them came from rural Ethiopia. From the study done in southwest Ethiopia, DDT and its metabolite DDE were detected in most of the staple food items such as teff, maize, and red pepper [12].

In our study, a higher mean concentration of total DDT was detected in the serum of cases than (5.20 ± 4.84) than controls (4.37 ± 2.82 μg/L). On the contrary, a study done in Belgium [28], revealed that, the total DDT concentration was lower both in cases (3.94 ± 3.88 μg/L) and controls (1.83 ± 1.98 μg/L). This variation might be due to the early banning of DDT from the market in Europe. From the DDT metabolites p.p'-DDE, o,p'-DDT, p,p'-DDD were not associated with increased risk of breast cancer in this study. Our result is supported by studies done in Copenhagen [29], in South African [30], and in Japan [31].

In the present study, a significantly (p<0.05) higher concentration of p,p'-DDT, and gamma chlordane were detected in breast cancer pataints than controls. In addition, p,p' DDT, and gamma chlordane were significantpredictors for increased risk of breast cancer. The result might indicate that one of the main risk factors for breast cancer in Ethiopia could be exposure to these two chemical pesticides. Similar findings have been reported by [26, 32, 33], where a positive association of gamma-chlordane and p,p'-DDT with increased risk of breast cancer.

From the analyzed pesticides, aldrin, endosulfan, endrin, dieldrin, dibutyl-chlorendate, and methoxychlor were also detected in serum of a few study participants. However, none of these pesticides showed a positive association with increased risk of breast cancer rather dibutyl-

chlorendate was found to be protective for breast cancer before adjustment ($P = 0.033$) (OR, 0.472, 95% CI, 0.236–0.942). This may be due to the less detection of these pesticides or maybe decreased use of these pesticides in the study area.

## Conclusions

Organochlorine pesticides (parents and metabolites) which were banned for agricultural use in most developed countries were detected in the serum of study participants in Ethiopia. This may be due to historical and recent uses of organochlorine pesticides in the country. Furthermore, the higher serum concentration and percent detection of heptachlor in is a good indicator for recent uses in our study area and exposure of the human population to these pesticides. Among the pesticides analyzed, p'p-DDT and gamma chlordane are an independent predicting factors for breast cancer development in Ethiopia and needs strict intervention. In general, the present study shows organochlorine pesticides are the potential risk factors for the development of a life-threatening breast cancer disease for women. Therefore, strict regulation on internationally banned pesticides and increase the awareness of the community toward the usage and effects of pesticides is essential to assure the safety of the community.

## Supporting information

**S1 Data.**
(DOCX)

**S1 Dataset.**
(SAV)

**S1 Questionnaire.**
(DOCX)

## Acknowledgments

We are highly indebted to the Organization for Women in Science for The Developing World (OWSD) Secretariat for their unreserved support and follow-up of the proper running of the project. We also would like to acknowledge Jimma University for giving us administrative support. We are grateful to the College of Health Science, Addis Ababa University, particularly, Oncology department for giving us permission to undertake the study. We are also very thankful for Sr. Mihret and Mr. Bersisa for supporting us in data as well as blood sample collection and serum separation.

## Author Contributions

**Conceptualization:** Seblework Mekonen, Mohammedgezali Ibrahim.

**Data curation:** Seblework Mekonen, Mohammedgezali Ibrahim, Higemengist Astatkie, Aynalem Abreha.

**Formal analysis:** Seblework Mekonen, Mohammedgezali Ibrahim, Aynalem Abreha.

**Funding acquisition:** Seblework Mekonen.

**Investigation:** Seblework Mekonen, Mohammedgezali Ibrahim, Aynalem Abreha.

**Methodology:** Seblework Mekonen, Mohammedgezali Ibrahim, Higemengist Astatkie.

**Project administration:** Seblework Mekonen.

**Resources:** Seblework Mekonen, Aynalem Abreha.

**Supervision:** Seblework Mekonen, Higemengist Astatkie, Aynalem Abreha.

**Validation:** Seblework Mekonen.

**Visualization:** Seblework Mekonen.

**Writing – original draft:** Seblework Mekonen, Mohammedgezali Ibrahim, Higemengist Astatkie.

**Writing – review & editing:** Seblework Mekonen, Mohammedgezali Ibrahim, Higemengist Astatkie, Aynalem Abreha.

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
