## [Decision Letter · Decision Letter 0]

17 Aug 2021

PONE-D-21-17256

Exposure to organochlorine pesticides as a predictor to breast cancer: a case-control study among Ethiopian women

PLOS ONE

Dear Dr. Mekonen,

Thank you for submitting your manuscript to PLOS ONE. After careful consideration, we feel that it has merit but does not fully meet PLOS ONE’s publication criteria as it currently stands. Therefore, we invite you to submit a revised version of the manuscript that addresses the points raised during the review process.

We look forward to receiving your revised manuscript.

Kind regards,

Swatantra Pratap Singh, Ph.D.

Academic Editor

PLOS ONE

Journal Requirements:

We are highly indebted to the Organization for Women in Science for The Developing World (OWSD) Secretatiate for their unreserved support and follow-up of the proper running of the project. This study was carried out by the aid of a grant obtained from UNESCO and International Development Research Cntre (IDRC) Ottawa Canada, and the authors deeply acknowledged the funders.

The grant holder is Seblework Mekonen,

The study was funded by a grant obtained from UNESCO/IDRC under the Organization for Women in Science for The Developing World (OWSD), Award agreement No.: 4500384863

The funders had no role in study design, data collection and analysis, decision to publish, or preparation of the manuscrip

6. We note that Figure 1 in your submission contain [map/satellite] images which may be copyrighted. All PLOS content is published under the Creative Commons Attribution License (CC BY 4.0), which means that the manuscript, images, and Supporting Information files will be freely available online, and any third party is permitted to access, download, copy, distribute, and use these materials in any way, even commercially, with proper attribution. For these reasons, we cannot publish previously copyrighted maps or satellite images created using proprietary data, such as Google software (Google Maps, Street View, and Earth). For more information, see our copyright guidelines: http://journals.plos.org/plosone/s/licenses-and-copyright.

Additional Editor Comments (if provided):

The manuscript require a minor revision for publication.

Reviewers' comments:

Reviewer's Responses to Questions

**Comments to the Author**

1. Is the manuscript technically sound, and do the data support the conclusions?

Reviewer #1: Yes

Reviewer #2: Yes

2. Has the statistical analysis been performed appropriately and rigorously? 

Reviewer #1: Yes

Reviewer #2: Yes

3. Have the authors made all data underlying the findings in their manuscript fully available?

Reviewer #1: Yes

Reviewer #2: No

4. Is the manuscript presented in an intelligible fashion and written in standard English?

Reviewer #1: Yes

Reviewer #2: Yes

5. Review Comments to the Author

Reviewer #1: This paper investigates the role of organochlorine pesticide as a predictor of breast cancer. In the study, authors have collected samples and analyzed for the pesticide levels. In addition, they have also conducted survey of the participants. Further, detailed statistical analyses are done to correlate the presence of pesticide to the breast cancer. The study is well designed. In addition, comprehensive analyses are performed and reported. I recommend that this manuscript could be accepted with minor revision. There are few issues which can be addressed, as mentioned below:

1. The language could be improved since there are many spelling and grammatical mistakes throughout the manuscript.

2. References are not properly cited - see 2nd line in introduction. Also, multiple reference styles are used - see page 6, second line.

3. Provide full form at first occurrence - OCP in last paragraph of introduction.

Reviewer #2: The article highlights a very important environmental issue. However, there has to be some revisions before acceptance

Page 3: Please rephrase the last sentence “The remaining unidentified causes continue to interest the scientific communities to gave attention to…”

In the introduction it is suggested to incorporate about a statistics/bar chart from literature as to which developed and developed countries are talked about

Please spell check the manuscript

Please use units uniformly.

Section 3.1: Report values only up to significant decimal values

In the abstract it says data was collected for 100 people, however the total comes out only to be 50.

Please grammar check the manuscript

For logistic regression please explain it terms of response and independent variables

Make sure that the independent variables are mutually exclusive, may not be in some cases

More detailed explanations of OR significance required, what does it mean it terms of behaviour of the response

Each acronym has to be expanded when first used

6. PLOS authors have the option to publish the peer review history of their article (what does this mean?). If published, this will include your full peer review and any attached files.

Reviewer #1: No

Reviewer #2: No

---

## [Author Response · Author response to Decision Letter 0]

31 Aug 2021

Dear Editor and reviewers of PLOS ONE journal, the authors are very much grateful for giving us the opportunities to review and submit the revised version of our manuscript and for your scholarly comments. All the comments are addressed in the manuscript and the major changes are shown in track change. All specific comments of editor and reviewers are well addressed and submitted with files as Response to Reviewers.

---

## [Editor Report · Decision Letter 1]

8 Sep 2021

Exposure to organochlorine pesticides as a predictor to breast cancer: a case-control study among Ethiopian women

PONE-D-21-17256R1

Dear Dr. Mekonen,

We’re pleased to inform you that your manuscript has been judged scientifically suitable for publication and will be formally accepted for publication once it meets all outstanding technical requirements.

Kind regards,

Swatantra Pratap Singh, Ph.D.

Academic Editor

PLOS ONE

---

## [Editor Report · Acceptance letter]

14 Sep 2021

PONE-D-21-17256R1 

Exposure to organochlorine pesticides as a predictor to breast cancer: a case-control study among Ethiopian women 

Dear Dr. Mekonen:

I'm pleased to inform you that your manuscript has been deemed suitable for publication in PLOS ONE. Congratulations! Your manuscript is now with our production department. 

Kind regards, 

on behalf of

Dr. Swatantra Pratap Singh 

Academic Editor

PLOS ONE